# New Insight on the Formation of 2-Aminoacetophenone in White Wines

Tatjana Radovanović Vukajlović [1], Christian Philipp [2], Phillip Eder [2], Martin Šala [3], Vid Simon Šelih [3], Andreja Vanzo [4], Katja Šuklje [4], Klemen Lisjak [4], Melita Sternad Lemut [1], Reinhard Eder [2] and Guillaume Antalick [1,*]

1. Wine Research Centre, University of Nova Gorica, Lanthieri Palace, Glavni trg 8, 5271 Vipava, Slovenia; tatjana.radovanovic@ung.si (T.R.V.); melita.sternad.lemut@ung.si (M.S.L.)
2. Federal College and Research Institute for Oenology and Pomology, Wienerstraße 74, 3400 Klosterneuburg, Austria; christian.philipp@weinobst.at (C.P.); phillip.eder@weinobst.at (P.E.); reinhard.eder@weinobst.at (R.E.)
3. National Institute of Chemistry, Hajdrihova 19, 1000 Ljubljana, Slovenia; martin.sala@ki.si (M.Š.); vid.selih@ki.si (V.S.Š.)
4. Department of Fruit Growing, Viticulture and Oenology, Agricultural Institute of Slovenia, Hacquetova Ulica 17, 1000 Ljubljana, Slovenia; andreja.vanzo@kis.si (A.V.); katja.suklje@kis.si (K.Š.); klemen.lisjak@kis.si (K.L.)
* Correspondence: guillaume.antalick@ung.si

**Abstract:** This study aimed to improve the understanding of 2-aminoacetophenone (2-AAP) formation in white wines, a compound responsible for atypical ageing (ATA) associated with a rapid loss of white wine fruity aroma and to the development of unpleasant odors. Two surveys performed in 139 white wines from Central Europe investigated the varietal effect on wine tendency to form 2-AAP during ageing. The role of some antioxidants and important transition metal ions present in wine (Fe, Cu and Mn) in the formation of 2-AAP was also investigated. The surveys highlighted that Muscat and Riesling wines showed significantly higher concentrations of 2-AAP than Chardonnay and some other regional varieties found in Central Europe such as Grüner Veltliner, Welschriesling and Zelen. The origin of such varietal effects could not be related to the level of any 2-AAP precursor. On the other hand, some of the important wine matrix effects were highlighted. A certain variability in capacity of different antioxidants to limit 2-AAP formation was observed. Supplementations of commercial tannins in wines tended to be more efficient than glutathione in preserving wines from the formation of 2-AAP. Transition metal ions significantly impacted 2-AAP synthesis through complex interactions. Generally, Fe and Mn tended to promote formation of 2-AAP, while the presence of Cu limited it. The higher concentration of transition metal ions significantly improved the efficiency of antioxidants to reduce 2-AAP formation. Further studies investigating the origin of varietal effects and the complex matrix effects involving metal ions and antioxidants in 2-AAP production are warranted.

**Keywords:** 2-aminoacetophenone; transition metal ions; antioxidants; atypical ageing

## 1. Introduction

Atypical ageing (ATA), also called untypical ageing (UTA), is an off-flavor appearing in young white wines, and it is associated with a rapid loss of a wine's fruity aroma and to the development of unpleasant odors such as acacia blossom, mothball, dirty dish or furniture varnish [1,2]. While ATA has been well-documented in the literature, its chemical characterization is still incomplete [2]. To date, 2-aminoacetophenone (2-AAP) has been identified as the main compound responsible for its occurrence in wines [1]. Its sensory threshold was reported to be from 0.5 µg/L to 10.5 µg/L according to the wine matrix [3]. Christoph et al. [4] described 2-AAP formation by radical co-oxidation of sulfite to sulfate, which leads to the oxidative degradation of 3-indoleacetic acid (3-IAA), a compound derived from tryptophan metabolism. Hoenicke et al. [5] proposed that the pyrrole ring

cleavage of 3-IAA by superoxide radicals, produced by the aerobic oxidation of sulfite during storage of sulfurized wines, form 3-(2-formylaminophenyl)-3-oxopropinoic acid (FAPOP). The decarboxylation of FAPOP leads afterwards to the formation of N-formyl-2-aminoacetophenone (FAP) and finally to 2-AAP, or alternatively, oxidized indole–acetic acid. The possibility that other metabolites such as tryptophan, indole-3-lactic acid (ILA) and kynurenine could be co-oxidized to 2-AAP after sulfite addition was also investigated, but these reactions only had a minor contribution to the formation of 2-AAP in wine [4]. The photooxidation of free and protein-bound tryptophan in combination with storage conditions could also potentially yield 2-AAP [6]. On the other hand, 2-AAP could be formed by yeasts during alcoholic fermentation, but again in concentrations below the sensory threshold [7,8].

While the chemical reactions yielding 2-AAP concentrations in white wines have been well described, the factors driving its formation in wines are still poorly understood [2]. The high levels of free 3-IAA at sulfiting increases the risk of obtaining higher levels of 2-AAP later in the wine, but no close correlation was established between the concentrations of 3-IAA and 2-AAP in wines [2]. The formation of 2-AAP requires only some residual traces of dissolved oxygen, which are incapable of leading to classical oxidative ageing. However, the presence of antioxidants before sulfiting was reported to prevent the formation of 2-AAP [4,9]. It has been shown that the addition of ascorbic acid could strongly inhibit the formation of 2-AAP [10]. However, the use of ascorbic acid in winemaking can promote the formation of strong oxidants if at the same time the level of sulfur dioxide is too low [11]. Consequently, research has been carried out to find an alternative to ascorbic acid in order to prevent the formation of 2-AAP in wine. Among other antioxidants used in oenology, the effect of tannin and glutathione supplementations in wine did not consistently limit the 2-AAP formation [2]. In contrast, Nardin et al. [3] recently showed that the addition of tannins in grape must prior to fermentation was more efficient than glutathione in preventing the formation of 2-AAP in wine, with gallotannin showing an effect almost comparable to ascorbic acid.

Another important factor to consider in wine oxidation chemistry is the role of transition metal ions. This has been well documented for some important chemical reactions like the oxidation of ethanol into acetaldehyde [12]. On the other hand, the impact of transition metal ions on ATA development has been only suggested by sensory study without any quantification of 2-AAP [13]. Finally, a preliminary study performed in our laboratory (data not published) suggested that some varieties might be more prone to the formation of 2-AAP than others. This was intriguing, as while historically, Riesling wines have been known to be particularly sensitive to ATA, this off-flavor was observed in different types of white wines all around the world [2].

This study aimed to improve the understanding of 2-AAP formation in white wines by investigating three main questions. Firstly, the potential varietal effect on 2-AAP concentration in white wines was assessed by analyzing two set of commercial white wines, and included the quantification of 2-AAP after artificial ageing and 2-AAP precursors. Secondly, the effect of different types of antioxidants on 2-AAP formation was investigated, as previously published results were unclear on the topic. Finally, the influence of transition metal ions present in wine (Fe, Cu and Mn) on 2-AAP concentration was performed for the first time.

## 2. Materials and Methods

L-tryptophan ($\leq$98%), 3-indoleacetic acid (98%), DL-indole-3-lactic acid (99%), DL-kynurenine ($\leq$95%), anthranilic acid ($\leq$98%), 2-aminoacetophenone (98%), glutathione (98%), iron (II) sulfate heptahydrate (99.5%), iron (III) chloride hexahydrate (97%), manganese (II) sulfate monohydrate (98%) and gallic acid (98%) were supplied by Sigma-Aldrich (Steinheim, Germany). LC-MS methanol was from Supelco (Bellefonte, PA, USA), and LC-MS formic acid was from Honeywell-Fluka (Charlotte, NC, USA). Copper (II) sulfate pentahydrate (98%) and high purity grade ethanol (>99.8%) was supplied by Honeywell-

Riedel-de Haën (Charlotte, NC, USA). 2-aminoacetophenone-D5 was supplied by EPTES (Vevey, Switzerland). Commercial ellagitannin (TA1), a commercial mixture of ellagitannin and condensed tannins (TA2), and inactivated yeast derivatives (IYD) were kindly provided by the Laffort company (Floirac, France). Chemicals were prepared with mili-Q water obtained from an ELGA PURELAB Option water system supplied by Elga-Veolia (Buckinghamshire, UK).

### 2.1. Surveys of 2-Aminoacetophenone in Commercial Varietal Wines

Two surveys of 2-AAP content in commercial varietal wines were carried out. The first survey consisted of analyzing 2-AAP concentration in 86 commercial Austrian white wines from the 2021 vintage prior to and after ageing at 40 °C for four days. This set of wines was sourced from producers and was exclusively composed of four varieties: Grüner Veltliner (20), Chardonnay (16), Riesling (25) and Muscat Ottonel (25) (Supplementary Data S1). The choice of the varieties was based on their importance for the Austrian wine market and by considering a preliminary study highlighting several trends of higher 2-AAP in some of the selected varieties. Ageing parameters were selected based on a recent study from Nardin et al. [3] showing that ageing different white varieties for 3 and 6 days at 40 °C was enough to substantially increase the level of 2-AAP. After 4 days of artificial ageing, samples were stored at 4 °C for a few days prior to analysis by gas chromatography coupled to a triple-quad mass detector (GC-MS/MS).

A second commercial white wine screening consisted of 45 wine samples from Slovenia (40) and Croatia (5) purchased from shops and producers. Samples from vintages 2018, 2019, 2020 and 2021 were analyzed in 2022 as in the first survey. Selected varieties were the same as for the Austrian survey, however, Grüner Veltliner was excluded (rarely found in Slovenia and Croatia) and replaced by the locally more important varieties of Zelen (Slovenia) and Welschriesling (Croatia) (Supplementary Data S1). The 2-AAP concentration was measured in these wines without artificial ageing.

In the same wines, the level of 2-AAP precursors tryptophan, anthranilic acid, indolelactic acid (ILA), 3-indoleacetic acid (3-IAA) and kynurenine were measured by liquid chromatograph coupled to a triple-quad mass detector (LC-MS/MS).

In order to investigate the influence of ILA on 2-AAP formation, 2-AAP was analyzed in a 2021 white blend from Slovenia (wine A) before and after artificial ageing for 4 days at 40 °C, and in the same wine spiked with 3-IAA and ILA at 100 µg/L after artificial ageing.

### 2.2. Experiments Using Antioxidants and Transition Metal Ions Supplementation
2.2.1. Antioxidants Addition

Two studies investigating the effect of antioxidants on 2-AAP formation were carried out in two white wines (A and B). Both wines were initially spiked with 100 µg/L of 3-IAA in order to ensure 2-AAP production. Chemical parameters of wines A and B are displayed in Supplementary Data S2.

Experiment 1 was performed in 2021 on a 2020 white blend from Slovenia (wine B). Initial free sulfite and 2-AAP concentrations were 25 mg/L and 0.75 µg/L, respectively. A series of different antioxidants was tested as follows: glutathione (GSH) at 12 mg/L, inactivated yeast derivatives (IYD) at 400 mg/L, gallic acid (GA) at 70 mg/L, commercial ellagitannins (TA1) at 100 mg/L, commercial mixture of ellagitannins and condensed tannins (TA2) at 100 mg/L and inactivated yeast derivatives (400 mg/L) in combination with commercial ellagitannins (100 mg/L). The targeted values of antioxidant additions were set according to supplier recommendations.

Experiment 2 was performed in 2022 on wine A again. Initial concentrations of free sulfite, 2-AAP, Cu, Mn and Fe were 22 mg/L, 0.35 µg/L, 75 µg/L, 796 µg/L and 1047 µg/L, respectively. A series of different antioxidants were tested as follows: GSH at 12 mg/L, inactivated yeast derivatives (IYD) at 400 mg/L, commercial ellagitannin (TA1) at 10 mg/L and 100 mg/L and inactivated yeast derivatives (400 mg/L) in combination with commercial ellagitannin (100 mg/L). Each antioxidant treatment was tested alone and

in combination with a mixture of metal ions to reach the final concentration as follows: Cu (II) at 1 mg/L using copper (II) sulfate pentahydrate solution; Fe(II)/Fe(III) (1:1) at 5 mg/L using iron (II) sulfate heptahydrate and iron (III) chloride hexahydrate solutions; and Mn (II) at 4 mg/L using manganese (II) sulfate monohydrate solution.

Each treatment was prepared in triplicate adding the antioxidant and metal ion solutions in 20 mL vials filled with 18 mL of wine and tightly sealed with PTFE-lined caps. Vials were then exposed to artificial ageing in the oven for 4 days at 40 °C. Thereafter, vials were stored in the fridge for a few days prior to analysis.

### 2.2.2. Metal Ions Addition

A different wine, a Slovenian white blend from 2021 (wine C), was used in 2022 to test the effect of transition metal ions alone and in mixture on the formation of 2-AAP in white wine. Initial concentrations for free sulfite, 2-AAP, Cu, Mn and Fe were 20 mg/L, 0.83 µg/L, 58 µg/L, 872 µg/L and 564 µg/L, respectively. Transition metals were tested using the same solution freshly prepared as in experiment 2. The following treatments were tested: Fe(II) at 5 mg/L; Fe(III) at 5 mg/L; Fe(II) and Fe(III) (1:1) at 5 mg/L; Fe (II/III) and Mn (II) at 5 mg/L and 4 mg/L, respectively; Cu (II) at 1 mg/L; Mn (II) at 4 mg/L; Cu (II) and Mn (II) at 1 mg/L and 4 mg/L, respectively; Fe(II/III) (1:1), Mn(II) and Cu(II) at 5 mg/L, 4 mg/L and 1 mg/L, respectively.

Each treatment was prepared in triplicate by adding metal ion solutions in 20 mL vials filled with 18 mL of wine and tightly sealed with PTFE-lined caps. Vials were further exposed to artificial ageing in the oven for 4 days at 40 °C. Thereafter, vials were stored in the fridge for a few days prior to analysis.

### 2.3. GC-MS/MS Measurement of 2-AAP

All wines were analyzed for the content of 2-AAP by HS-SPME-GC-MS/MS. For this purpose, 5 mL of mQ water, 5 mL of sample, 50 µL of 2-AAP-d5 solution prepared at 500 µg/L in high purity grade ethanol and 3 g of NaCl were added to a 20 mL SPME vial. Analysis was performed using a gas chromatograph (type 7890 B A, Agilent Technologies, Santa Clara, CA, USA) with an injector, a controller, a CTC Analytics autosampler (Zwingen, Switzerland) and a triple-quad mass spectrometer (MS/MS) detector (type 7010B GC/MS Triple Quad) (Agilent Technologies, Santa Clara, CA, USA). Separation was performed using a ZB-5MS column (30 m × 0.25 mm I.D. × 0.25 µm df; Agilent Technologies, Santa Clara, CA, USA) at a column flow rate of 1.02 mL/min. Helium was used as the carrier gas. The blue fiber coated with polydimethylsiloxane–divinylbenzene 65 µm (PDMS-DVB) (Supelco, Bellefonte, PA, USA) was used with the following settings: pre-incubation time: 60 s, incubation temp.: 70 °C, pre-incubation agitator speed: 500 rpm, agitator-on time: 5 s, agitator-off time: 2 s, fiber exposure: 12 µL, vial penetration: 15 mm, extraction time: 2400 s, injection penetration: 32 mm, desorption time: 120 s, fib. cond. temp.: 250 °C, and post-fib. cond. time: 600 s. The inlet was set in splitless mode at 250 °C. The oven temperature was programmed at 50 °C for 2 min before a rise of 15 °C/min to reach 160 °C, which was held for 1 min. Then, the temperature was increased by 20 °C/min to reach 230 °C and held for 5 min. The postrun temperature was set at 60 °C. The total run time was 19 min. The transfer line temperature was set at 250 °C. The mass spectrometer was operated in electron ionization (EI) mode at 70 eV with multiple reaction monitoring (MRM). All of the monitored transitions are listed in Table 1. Data were acquired and analyzed through Agilent MassHunter Workstation software, version B.07.00. The calibration was performed in 8 calibration steps in the range from 0 to 9.6 µg/L in the synthetic wine ($R^2$ = 0.9993). Limit of detection and limit of quantification were calculated at 0.1 and 0.3 µg/L by considering signal to noise S/N = 3 and S/N = 10, respectively. Recovery was assessed at 99% by spiking one-year old white wine at 0.6 µg/L.

**Table 1.** Mass spectral transitions and collision energies selected for GC-MS/MS analysis of 2-AAP-d5 and 2-AAP.

| Compounds | Retention Time (min) | First Transition (Quantifier) | | Second Transition (Qualifier) | |
| | | Precursor Ion to Product Ion (*m/z*) | Collision Energy (eV) | Precursor Ion to Product Ion (*m/z*) | Collision Energy (eV) |
|---|---|---|---|---|---|
| 2-AAP-d5 | 9.33 | 140 → 122 | 12 | 140 → 94 | 10 |
| 2-AAP | 9.37 | 135 → 120 | 12 | 135 → 92 | 10 |

*2.4. ICP/MS Measurement of Metal Ions*

Transition metal ions were analyzed by ICP-MS. All reagents used were of analytical grade or better. For sample dilution and preparation of standards, ultrapure water (MilliQ, Millipore) and ultrapure acids ($HNO_3$ and HCl, Merck-Suprapure) were used. Standards were prepared in-house by dilution of certified, traceable, inductively coupled plasma (ICP)-grade single-element standards (Merck CertiPUR). The standards and blanks were prepared in model wine consisting of 12% ethanol, 4 g/L glucose and 4 g/L tartaric acid, with the pH adjusted to 3.4. An Agilent Technologies 7900 ICP-MS instrument equipped with a MicroMist glass concentric nebulizer and a Peltier-cooled, Scott type spray chamber was used.

*2.5. UHPLC-MS/MS Measurement of 2-AAP Precursors*

Analysis was carried out by a 1290 infinity UHPLC system coupled to a 6460 triple-quadrupole mass spectrometer (Agilent Technologies). Wines were diluted in methanol (1:1, *v/v*), vortexed, filtered through a 0.22 μm PVDF filter (Millipore, Billerica, MA, USA) and directly injected. Samples were kept at 4 °C during analysis, and the injection volume was 5 μL. Separation was performed on a 100 mm × 2.1 mm, 1.8 μm column (Acquity HSS T3, Waters) maintained at 40 °C. Flow was 0.4 mL/min; mobile phase A was 0.1% formic acid and B was 0.1% formic acid in methanol. The linear gradient started at 2% B, to 55% B in 45 min, to 100% B in 1 min, at 100% B for 3 min, to 2% B in 1 min and then 5 min at 2% B. Analysis was performed in positive mode using electrospray ionization (ESI, Jet-Stream) and selected reaction monitoring (SRM) acquisition. Nitrogen was the collision gas, and the source parameters were gas temperature 250 °C, gas flow 6 L min$^{-1}$, nebulizer 241 Kpa, sheath gas heater 375 °C, sheath gas flow 10 L min$^{-1}$, capillary 4000 V. Three mass transitions were selected for each compound: one for the quantifier and two for qualifiers (Table 2). The linearity of the detector response for 2-AAP precursors was verified in synthetic wine diluted in methanol (1:1, *v/v*). Seven concentration levels were injected in six repetitions for each level. Linearity was determined, and the limit of detection and limit of quantification were calculated by considering signal to noise S/N = 3 and S/N = 10, respectively (Table 2).

**Table 2.** Chromatographic retention times, selected reaction monitoring conditions (Q-quantifier, q-qualifiers), linear regression (R2), limit of detection (LOD) and limit of quantification (LOQ) for 2-AAP precursors in wines.

| Compound | Retention Time (min) | Parent (*m/z*) | Q (*m/z*) | q (*m/z*) | q (*m/z*) | Concentration Range | R$^2$ | LOD | LOQ |
|---|---|---|---|---|---|---|---|---|---|
| Tryptophan (mg/L) | 5.7 | 205.1 | 146.1 | 118.1, | 91.1 | 0.1–10 | 0.99974 | 0.01 | 0.04 |
| Kynurenine (μg/L) | 3.2 | 209.1 | 146.0 | 94.1 | 65.1 | 0.5–10 | 0.99736 | 0.23 | 0.56 |
| Anthranilic acid (μg/L) | 7.6 | 138.1 | 65.1 | 92.0 | 39.1 | 0.1–40 | 0.99955 | 1.20 | 1.33 |

**Table 2.** *Cont.*

| Compound | Retention Time (min) | Parent (*m/z*) | Q (*m/z*) | q (*m/z*) | q (*m/z*) | Concentration Range | $R^2$ | LOD | LOQ |
|---|---|---|---|---|---|---|---|---|---|
| ILA (µg/L) | 11.1 | 206.1 | 160.0 | 130.0, | 118.1 | 0.5–245 | 0.99918 | 2.50 | 7.37 |
| 3-IAA (µg/L) | 12.8 | 176.1 | 130.0 | 103.0 | 77.1 | 0.5–245 | 0.99953 | 0.67 | 1.48 |

*2.6. Statistical Analysis*

Comparisons between treatments were performed by one-way ANOVA (analysis of variance) using Microsoft Excel 2016. The threshold for statistical significance was set at 95% (*p*-value < 0.05). Box plots were also made using Microsoft Excel 2016 software.

**3. Results**

*3.1. Surveys of 2-Aminoacetophenone in Commercial Wines*

The first survey aimed to measure 2-AAP concentrations in 86 Austrian white wines from the 2021 vintage prior to and after ageing for 4 days at 40 °C in order to assess the potential of the same wines to produce 2-AAP after prolonged ageing. Results show that the level of 2-AAP concentrations were clearly influenced by the variety (Figure 1).

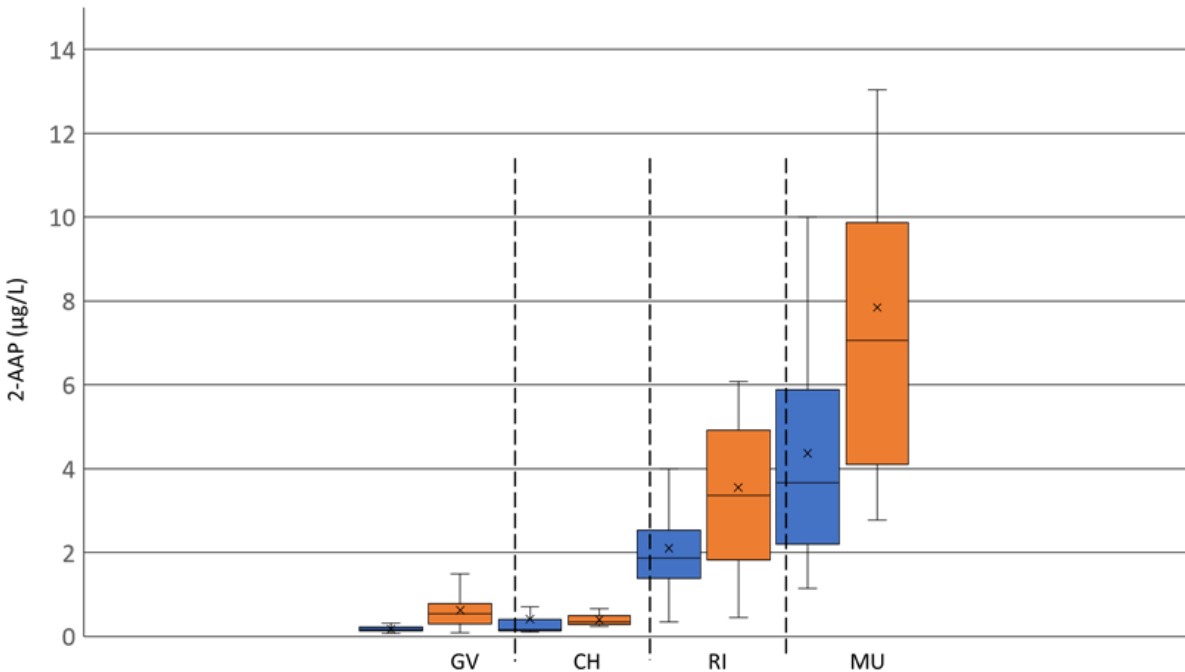

**Figure 1.** Box plot of 2-AAP concentration measured in 86 fresh and aged Austrian wines from four different varieties: GV—Grüner Veltliner (20), CH—Chardonnay (16), RI—Riesling (25) and MU—Muscat Ottonel (25) (blue color = wine before artificial ageing, orange color = aged wine). The crosses indicate the average concentrations.

The lowest 2-AAP concentrations before artificial ageing were measured in Grüner Veltliner and Chardonnay wines with average values at 0.17 µg/L and 0.27 µg/L, respectively. The highest average concentrations were obtained in Riesling and Muscat wines (2.3 and 4.3 µg/L, respectively). The maximum 2-AAP concentrations were reached in Muscat wines with some values up to 10 µg/L. The same varietal effect was observed after artificial ageing, which increased 2-AAP concentrations as expected. The highest gains of concentration were observed in Riesling and Muscat wines, with average gains of 1.1 µg/L and 3.5 µg/L, respectively. The final 2-AAP concentration after ageing increased

up to 20 μg/L in one Muscat wine. The lowest average increase in 2-AAP concentration after ageing was measured in Chardonnay wines (0.17 μg/L). On the other hand, the relative gain in concentration was the highest in Grüner Veltliner wines, with average concentrations 3.6-fold higher after than before ageing. Finally, the range of 2-AAP concentrations measured in the different samples were generally wider after ageing than in fresh wine. This was particularly the case for Grüner Veltliner, and, to a lesser extent, Chardonnay and Riesling wines.

The second survey aimed to measure 2-AAP and tryptophan metabolites considered as potential 2-AAP precursors in 53 Slovenian and Croatian white wines. For each variety, 2-AAP average concentrations were overall higher in this set of wines in comparison to the first survey (Table 3). Similar trends were observed, with 2-AAP contents substantially higher in Muscat and Riesling in comparison to Chardonnay wines. The lowest 2-AAP contents were measured in Zelen and Welschriesling wines. In contrast, Zelen wines show the highest average concentration in 3-indoleacetic acid (18.8 μg/L), which is known to be the main precursor to 2-AAP formation in wine. There was generally no correlation between 2-AAP concentration and the content of tryptophan metabolites. However, it can be noticed that ILA and anthranilic acids were significantly higher in Riesling wines, which often displayed a high level of 2-AAP. Finally, indole-3-lactic acid was by far the most abundant tryptophan metabolite, ranging from 50 μg/L to 350 μg/L, whereas 3-indoleacetic acid content ranged from 0.2 μg/L to 60 μg/L. ANOVA analysis a showed statistically significant difference in concentrations of anthranilic acid, 3-IAA, kynurenine and 2-AAP between five varietal wines.

**Table 3.** Concentration of 2-AAP precursors and 2-AAP (μg/L) determined in five Slovenian and Croatian varietal white wines (*n* = 45). One-way ANOVA was used to compare data. For each line, different letters represent significantly ($p \leq 0.05$) different concentrations between varieties.

| Variety | Chardonnay (*n* = 8) | Riesling (*n* = 10) | Muscat (*n* = 12) | Zelen (*n* = 9) | Welschriesling (*n* = 6) |
|---|---|---|---|---|---|
| Tryptophan | 1300 ± 700 a | 700 ± 700 b | 800 ± 700 b | 900 ± 700 b | 900 ± 700 b |
| Anthranilic acid | 7.3 ± 2.5 ab | 11.9 ± 6.4 a | 5.7 ± 2.9 b | 5.2 ± 3.0 b | 5.3 ± 3.1 b |
| ILA | 149 ± 59 b | 196 ± 97 a | 134 ± 109 b | 122 ± 79 b | 84 ± 23 c |
| 3-IAA | 2.2 ± 1.1 b | 4.5 ± 3.6 b | 4.8 ± 5.0 b | 18.8 ± 18.3 a | 3.3 ± 2.6 b |
| Kynurenine | 1.9 ± 1.1 a | 1.0 ± 0.7 b | 1.1 ± 0.6 b | 1.2 ± 0.6 b | 2.3 ± 1.8 a |
| 2-AAP | 1.2 ± 2.1 c | 4.2 ± 2.4 b | 8.7 ± 5.5 a | 0.3 ± 0.3 c | 0.3 ± 0.3 c |

### 3.2. Influence of 3-IAA and ILA on 2-AAP Concentration

In order to investigate the role of ILA on the formation of 2-AAP in white wines, a white blend was spiked with 3-IAA and ILA and heated for 4 days at 40 °C. Figure 2 shows that the addition of ILA did not yield more 2-AAP than the control treatment after ageing. Only the addition of 3-IAA led to the synthesis of higher contents of 2-AAP. Therefore, it seems that ILA does not yield 2-AAP in wine conditions.

### 3.3. Influence of Antioxidants

Another white blend (wine B) was spiked before artificial ageing with 3-IAA and different type of antioxidants including pure compounds (glutathione and gallic acid) and some oenological products (commercial tannins and inactivated yeast derivates). Concentrations of 2-AAP were measured in treated wines after artificial ageing.

Figure 3 shows that all tested antioxidants significantly decreased the formation of 2-AAP. However, the efficiency levels differed between treatments. The smallest decreases were observed with the addition of GSH and inactivated yeast derivates (IYD), whereas gallic acid and both types of commercial tannins were the most efficient at reducing the

formation of 2-AAP. The addition of IYD in combination with commercial ellagitannins did not modify 2-AAP concentration in comparison to the addition of tannins alone (Figure 3).

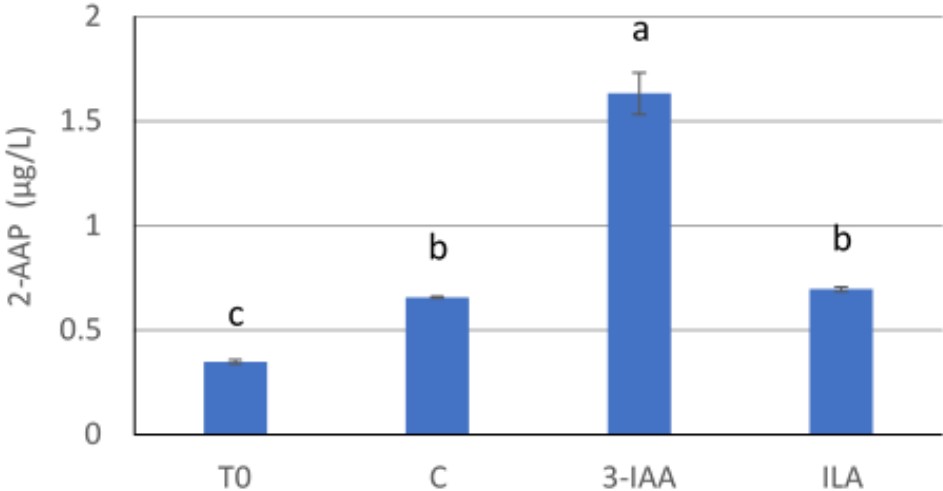

**Figure 2.** 2-AAP concentration measured in wine A before artificial ageing (T0) and after artificial ageing without addition (C) and with addition of 3-indoleacetic acid (3-IAA) and indole-3-lactic acid (ILA), both at concentration 100 μg/L. Different letters above bars indicate significance at $p \leq 0.05$ (Fischer's LSD). All quoted uncertainty is the standard deviation of three replicates per treatment.

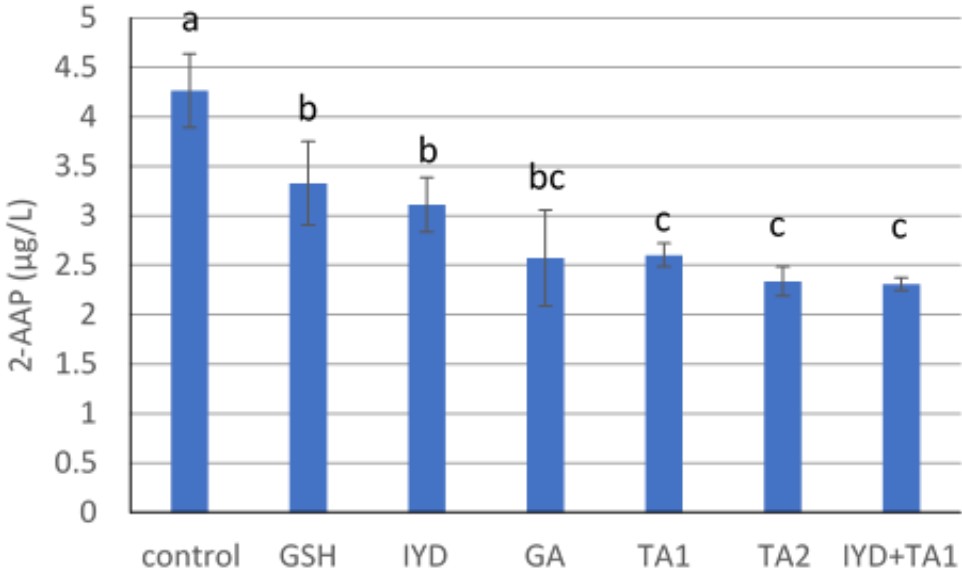

**Figure 3.** 2-AAP concentration measured in wine treated with different antioxidants (GSH—glutathione; IYD—inactivated yeast derivates, GA—gallic acid, TA1 and TA 2—different commercial tannins). Different letters above bars indicate significance at $p \leq 0.05$ (Fischer's LSD). All quoted uncertainty is the standard deviation of three replicates per treatment.

A second experiment consisted of testing the effects in wine A on the formation of 2-AAP from additions of two different concentrations of the antioxidants GSH, IYD and a commercial ellagitannin alone and in combination with IYD. Each antioxidant treatment was tested alone and with the addition of a mix of metal ions (Cu, Fe and Mn). The effect of antioxidants on the formation of 2-AAP was less significant in comparison to the first experiment. In this second wine, the addition of GSH and IYD did not modify the final concentration of 2-AAP (Figure 4). Only the addition of commercial ellagitannin (TA1) significantly reduced the 2-AAP concentration after artificial ageing. This decrease was more important with a higher dosage of tannins. Once again, the addition of IYD

did not modify the effect of tannin on the final level of 2-AAP, irrespective of tannin concentrations (Figure 4). When the metal ion mixture was added, the wines with GSH, IYD and tannin addition showed significantly lower levels of 2-AAP compared to the control wine (Figure 4). This decrease was the most significant with the higher dose of commercial tannin and the least with GSH. In those conditions, the addition of IYD was more efficient at reducing 2-AAP than GSH and as efficient as the low dosage of TA1. Interestingly, when IYD was combined with TA1, the 2-AAP concentration was the same as in the control wine. Finally, it should be noted that the addition of the metal ions mixture did not significantly modify 2-AAP content in the control wine. Nevertheless, the addition of the metal ion mixture led to a significant decrease in 2-AAP concentration after artificial ageing for each antioxidant treatment, excluding the one with a low dosage of tannins.

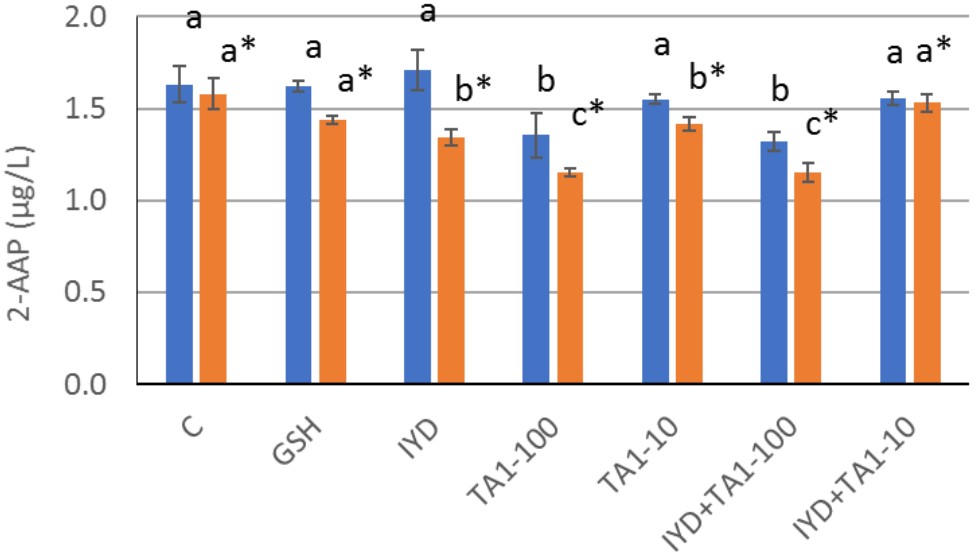

**Figure 4.** Treatment of wine with different antioxidants (blue—no metal ion addition, orange—metal ions addition; C—control, GSH—glutathione, IYD—inactivated yeast derivates, TA1—commercial ellagitannin). Metal ions were added to reach 1 mg/L, 5 mg/L and 4 mg/L of Cu, Fe and Mn, respectively. Different letters above bars indicate significance at $p \leq 0.05$ (Fischer's LSD). All quoted uncertainty is the standard deviation of three replicates per treatment.

### 3.4. Influence of Copper, Iron and Manganese

A white blend was spiked with salts of Cu(II), Fe(II), Fe(III) and Mn(II) in order to investigate the effect of these metal ions on the formation of 2-AAP. The additions were made with individual metal ions and with various mixtures of metal ions in order to reach their high concentration range found in wine [14]. In that respect, the targeted final concentration was 1 mg/L, 5 mg/L and 4 mg/L for Cu, Fe and Mn, respectively. Figure 5 shows that addition of Fe(II), Fe(III) and Mn(II) increased 2-AAP concentration in comparison to the control wine, with the highest content found in the wines spiked with Mn(II) alone. On the other hand, the addition of Cu(II) led to a significant decrease in 2-AAP. Interestingly, when Cu(II) was added in a mixture with other metal ions, the level of 2-AAP was systematically reduced (Figure 5). On the contrary, the addition of Mn(II) tended to increase the 2-AAP concentration in all the treatments, excluding when it was added with a mixture of Fe(II) and Fe(III). Finally, the addition of all the metal ions together led to a decrease in 2-AAP concentration after artificial ageing in comparison to the control treatment (Figure 5).

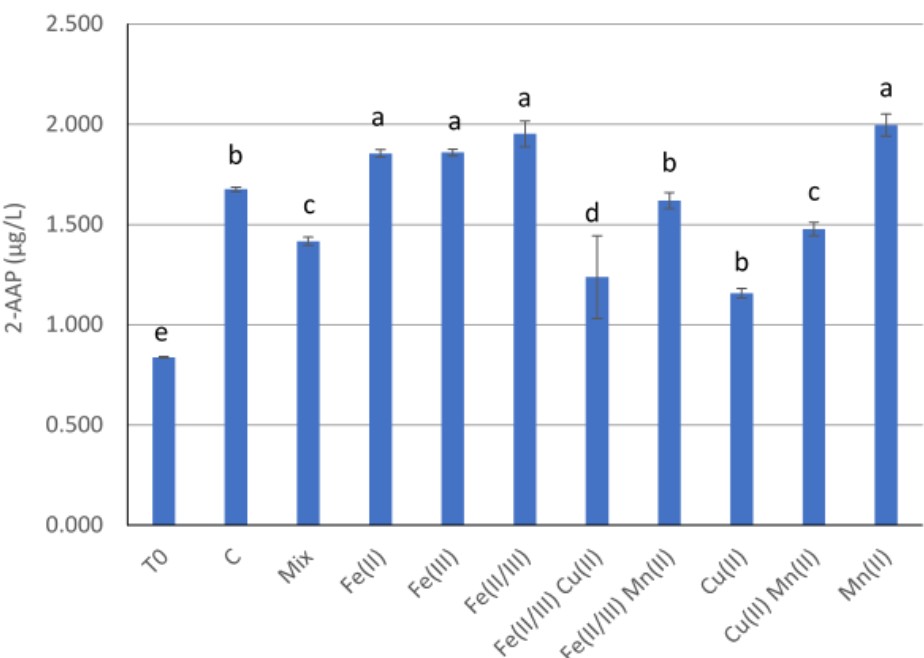

**Figure 5.** Levels of 2-AAP measured in wine before heat treatment and with application of different combination of metal ions (T0—control wine not exposed to heat treatment; C—wine exposed to heat treatment; MIX—mixture of Fe, Cu and Mn). Metal ions were added to reach 1 mg/L, 5 mg/L and 4 mg/L of Cu, Fe and Mn, respectively. Different letters above bars indicate significance at $p \leq 0.05$ (Fischer's LSD). All quoted uncertainty is the standard deviation of three replicates per treatment.

## 4. Discussion

Since the first empirical evidence of the development of ATA in white wines in the late 1980s, most of the published studies investigating the formation of 2-AAP in wine were carried out on Riesling [2]. This anecdotal observation suggests that the Riesling variety has been known for a long time to be sensitive to the formation of 2-AAP. On the other hand, very few studies have investigated a potential varietal effect on 2-AAP concentration in wine. Košmerl and Zlatić [15] analyzed 21 Slovenian wines, not including Riesling, and more recently, Alpeza et al. [16] measured 2-AAP in 20 Croatian white wines from different varieties, including Riesling. While the latter study found that the highest concentrations of 2-AAP were measured in Riesling wines, it is difficult to highlight a consistent trend with such a low number of samples. With two distinctive surveys representing a total of 139 wines analyzed, the present study highlighted that Riesling wines significantly showed a higher concentration of 2-AAP in comparison to Chardonnay and some other local varieties from Central Europe such as Grüner Veltliner, Welschriesling and Zelen. The average 2-AAP concentration in Riesling wines was between 2 and 4 µg/L. This is above the perception threshold reported for white wines (0.5–1.5 µg/L) [2]. However, the variety that displayed the highest average concentration of 2-AAP was Muscat blanc, and that observation was somewhat unexpected since ATA has been rarely reported in Muscat wines. This is probably due to the presence of a high concentration of monoterpenols imparting an intense floral aroma, which strongly mitigates the impact of 2-AAP on Muscat wine aroma [12].

The wider distribution of 2-AAP concentrations after ageing for most of the varieties indicated an important matrix effect in the formation of this compound in white wines. In order to clarify the matrix effect and varietal differences on 2-AAP formation, the second survey, performed on Slovenian and Croatian wines, targeted free tryptophan metabolites known as potential precursors of 2-AAP. No relationship between the level of 2-AAP and its precursors could be observed in the wines analyzed in this survey, including the main known precursor 3-indoleacetic acid (3-IAA). The lack of correlation between free 3-IAA

and 2-AAP in wines has already been reported by several authors [8,9,17]. Levels of 2-AAP potential precursors were in accordance with previously published concentrations [18]. The most abundant tryptophan derivate was not 3-IAA but indolelactic acid (ILA). Interestingly, Riesling wines showed the highest ILA concentrations in comparison to other varieties. However, when ILA was added to a white blend, the concentration of 2-AAP after artificial ageing was similar to the same wine without ILA addition. Only the wine with an addition of 3-IAA showed a significantly higher level of 2-AAP (Figure 2). Therefore, the varietal differences observed in 2-AAP concentration in white wines were not directly linked to the levels of its precursors in wine. Further investigation on varietal factors that could be directly or indirectly involved in 2-AAP formation are needed.

Moreover, Lavigne-Cruege et al. [19] suggested that glutathione and other yeast biomass derivates could limit 2-AAP formation, but this observation was not consistent with some other studies [20,21]. In the present work, phenolic compounds such as gallic acid and commercial tannins were more efficient at limiting the formation of 2-AAP than GSH and IYDs. While commercial tannins reduced the concentration of 2-AAP in both trials, GSH and IYDs had an impact only on one wine. Nardin et al. [3] recently published similar results using GSH, ellagic tannins, galla and grape tannins. They found that galla and grape tannins were the most efficient at limiting the 2-AAP concentration in the wine and that ellagic tannins had a stronger impact than GSH. In the present study, the mixture of grape and ellagic tannins (TA2) was also slightly more efficient than pure ellagic tannins (TA1). However, Nardin et al. [3] added antioxidants in must at higher levels in comparison to the present study, where the doses reflect additions suggested by manufacturers to be used in white wine. The addition of tannins in white wine at the upper limit of the recommended range often strongly impacts wine sensory profile. For that reason, our second trial targeted two levels of tannins addition corresponding to 10 g/hL (100 mg/L) and 1 g/hL (10 mg/L), respectively. The higher dosage was the most efficient to reduce 2-AAP concentration.

All these results suggest that when wines are aged in a barrel on gross lees, the tannins extracted from wood might be more efficient to limit the formation of 2-AAP than the antioxidant activity from the lees. Some synergistic effect between both classes of antioxidants cannot be excluded in some wines, as observed during the first trial where the addition of IYD with TA1 was slightly more efficient at reducing 2-AAP than TA1 alone (Figure 3). However, such an effect was not observed in the second wine, irrespective of tannin dosage (Figure 4).

Transition metal ions are known to catalyze many chemical reactions, but in a winelike medium, mainly Fe, Cu and, to a lesser extent, Mn have been studied for their role in wine oxidation and reduction [12,22]. Hoenicke et al. [5] speculated that transition metals could potentially influence the formation of 2-AAP in wine without studying this hypothesis. Morozova et al. [13] reported higher ATA perception in Riesling wines due to the presence of Fe and Cu but 2-AAP was not quantified. The present study is therefore the first to our knowledge to explore the impact of metal ions on 2-AAP production in wine. In the same preliminary study as mentioned earlier, a 2-fold increase in 2-AAP concentration (0.22 to 0.45 µg/L) was observed in the Chardonnay wine aged in an inox tank after the addition of a mixture of Fe, Cu and Mn in order to reach a final concentration at 5, 1 and 4 mg/L, respectively. In contrast, the 2-AAP level did not change when the same addition was made in the wine aged on gross lees in a new oak barrel. This observation suggests that antioxidants from oak or lees could also limit the influence of metal ions in the formation of 2-AAP.

When Fe, Cu and Mn were added together at the same level in two white blends, 2-AAP concentration after artificial ageing was not affected in the first wine (Figure 4) and was decreased in the second wine (Figure 5). Interestingly, the presence of higher concentrations of Fe, Cu and Mn in a mixture significantly improved the efficiency of antioxidants to reduce the 2-AAP concentration, while no metal ion effect was observed in the wines without antioxidant addition. Commercial tannins remained the most efficient



antioxidant to limit 2-AAP concentration in the presence of metal ions. These results highlight some complex interaction between metal ions and wine matrix involved in the formation of 2-AAP. Metal ions' reactivity is closely dependent on their redox potential, which is significantly influenced by the presence of ligands such as tartaric acid, malic acid, polyphenols or other antioxidants [23]. Fe(III/II) redox potential is also influenced by the presence of Cu and Mn [22,24]. Copper accelerates Fe(II) oxidation, and that in turn can promote sulfite oxidation [23,25], which is suspected to form the superoxide radical responsible for 2-AAP formation in white wines [5]. In contrast, in our experiment, the addition of Cu(II) tended to decrease the 2-AAP concentration. On the other hand, Mn was proven to be a strong promoter of sulfite oxidation in the presence of traces of Fe and Cu [22]. In our conditions, the presence of a higher level of Mn was actually rather associated with a higher content of 2-AAP. However, this trend was not observed when the wine had a higher initial Fe(II/III) concentration. This shows the complexity of chemical reactions yielding 2-AAP in a wine medium and can explain why the main limiting factors of 2-AAP formation in wine are not well identified yet. Further research will have to investigate more in detail the interactions between transitional metal ions and wine matrix.

This study opened new perspectives in the understanding of 2-AAP synthesis in white wines. Both varietal differences in 2-AAP formation and interactions between antioxidants and transition metals require further investigations.

**Supplementary Materials:** The following supporting information can be downloaded at: https://www.mdpi.com/article/10.3390/app13148472/s1, Table S1: Wine list for survey 1 and 2 with corresponding concentration in 2-AAP; Table S2: Supplementary data 2: Chemical parameter of wines used for the experiments.

**Author Contributions:** Conceptualization, G.A.; methodology, G.A., T.R.V., C.P., M.Š., V.S.Š. and A.V.; software, T.R.V., P.E., A.V., K.Š. and M.Š.; validation, T.R.V., P.E., C.P., M.Š., V.S.Š., A.V. and K.Š.; formal analysis, T.R.V., P.E., M.Š., A.V. and K.Š.; investigation, G.A., T.R.V., C.P., M.Š., V.S.Š., A.V. and K.L.; resources, G.A., C.P., K.L., M.S.L. and R.E.; data curation, G.A., T.R.V., P.E., C.P., M.Š., A.V. and K.Š.; writing—original draft preparation, T.R.V., G.A., C.P., M.Š. and A.V.; writing—review and editing, C.P., M.Š., V.S.Š., A.V., K.Š., K.L. and M.S.L.; project administration, M.S.L., G.A., R.E., A.V. and V.S.Š.; funding acquisition, G.A. All authors have read and agreed to the published version of the manuscript.

**Funding:** This work was cofunded by the Slovenian Research Agency (ARRS) research project L4-1842 titled "Influence of heavy metal ions on shelf life of white wines" and research program P1-0034 titled "Analysis and chemical characterization of materials and processes" and the Biolaffort company.

**Institutional Review Board Statement:** Not applicable.

**Informed Consent Statement:** Not applicable.

**Data Availability Statement:** The data presented in this study are available on request from the corresponding author.

**Acknowledgments:** We would like to thank Virginie Moine and Arnaud Massot from Biolaffort, and Nicolas Neve and François Botton from Laffort, for the scientific and technical discussions during this project. We would like also to acknowledge all the producers who provided wine samples for this work.

**Conflicts of Interest:** The authors declare no conflict of interest.

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
