# Peer review of "New Insight on the Formation of 2-Aminoacetophenone in White Wines"

_applsci, doi:10.3390/app13148472_

Round 1

Reviewer 1 Report

The work concerns the investigation of some aspects potentially related to the formation of 2-AAP in wines.

Initially, the study provides an important overview of the levels of 2-AAP in commercial wines of different varieties and countries, also evaluating its formation after artificial aging. In my opinion this part is very interesting also considering the high number of samples analysed, and if possible it should be given more attention. For example, also evaluating the vintage, or if the wines have been in barrels, etc...

The discussion is adequate to the observations and data obtained, is clear and opens the door for further research.

The text should be reviewed carefully due to the presence of typos and some inaccuracies

Line 90. Please provide company location (city, State)

Line 107. GC-MS/MS

Line 120. 40 °C

Line 168. 2-AAP

Line 169. How is preapared the internal standard? In Ethanol?

Line 176. Is the GC column flow rate really at 34.02 mL/min? It is very high both because it generates a pressure in the column head higher than the possibilities of the GC pneumatics and because it would limit the efficiency of the mass spectrometer: vacuum pumping difficulty, low ionization efficiency

Lines 183-185. Why is the pound sign with a number reported? In my opinion, the programmed temperature written in this way is not very clear

Lines 185-188. For better understanding, report the transitions with the respective collision energies and retention time in a table

Line 190. (R2 =

Line 192 and 223.  When the quantification is done using a triple quadrupole in MRM experiments it would be advisable to calculate LOQ and LOD based not on the S/N ratio because the background noise could be absent, furthermore noise can be modify by different smoothing treatment. It would be better calculate LOD and LOQ baseing on calibration slope and standard deviation (Evard et al., Analytica Chimica Acta, 2016,  942, 23-39.)

Line 198. HNO3

Line 211. 1.8 µm

Line 216. Carrier gas?

Line 217, 218. min-1

Line 219. Table X?

chapter 2.6. You should include what software the boxplots were made in

Table 1. m/z in italic

Figure 1. the two digits after the decimal point on the y-axis can be eliminated

Lines 243-256. Perhaps it could be useful to contextualize the data with respect to the olfactory threshold?

Line 287. Was wine A used for experiment 3.2? How were the two wines different? Some details could be added in supplementary

Line 418. Metal seizure by lees absorption?

Author Response

The answers to reviewer are written in green.

The work concerns the investigation of some aspects potentially related to the formation of 2-AAP in wines.

Initially, the study provides an important overview of the levels of 2-AAP in commercial wines of different varieties and countries, also evaluating its formation after artificial aging. In my opinion this part is very interesting also considering the high number of samples analysed, and if possible it should be given more attention. For example, also evaluating the vintage, or if the wines have been in barrels, etc...

We would like to thank reviewer 1 for the positive feedback and constructive comments he made. We added some information about geographical origin of wines analysed in the surveys in supplementary data 1. As always with such a survey of commercial wines it is difficult to have accurate technical information on all the wines. That’s why we did not include more information but that is a current practice in the literature for the reason explained above.

The discussion is adequate to the observations and data obtained, is clear and opens the door for further research.

The text should be reviewed carefully due to the presence of typos and some inaccuracies

Reviewer 1 is right; we went through to correct all the typos we found.

Line 90. Please provide company location (city, State)

Missing information about supplier was provided in material and method

Line 107. GC-MS/MS

This was corrected

Line 120. 40 °C

This was corrected

Line 168. 2-AAP

This was corrected

Line 169. How is preapared the internal standard? In Ethanol?

Yes, it was prepared in high purity grade ethanol. The information was added (line 185-186 in the revised manuscript)

Line 176. Is the GC column flow rate really at 34.02 mL/min? It is very high both because it generates a pressure in the column head higher than the possibilities of the GC pneumatics and because it would limit the efficiency of the mass spectrometer: vacuum pumping difficulty, low ionization efficiency

It was a mistake from our side. The right value is 1.02 mL/min. This was corrected

Lines 183-185. Why is the pound sign with a number reported? In my opinion, the programmed temperature written in this way is not very clear

We modified this part according to reviewer 1’s suggestion (line 199-201 in revised manuscript)

Lines 185-188. For better understanding, report the transitions with the respective collision energies and retention time in a table

We modified this part according to reviewer 1’s suggestion adding Table 1 in the revised manuscript

Line 190. (R2 =

This was corrected

Line 192 and 223.  When the quantification is done using a triple quadrupole in MRM experiments it would be advisable to calculate LOQ and LOD based not on the S/N ratio because the background noise could be absent, furthermore noise can be modify by different smoothing treatment. It would be better calculate LOD and LOQ baseing on calibration slope and standard deviation (Evard et al., Analytica Chimica Acta, 2016,  942, 23-39.)

This is a relevant comment indeed. Nevertheless, we considered that the noise of chromatograms was usable and that we could use the S/N approach to calculate LOD and LOQ. This approach is commonly used for GC-MS/MS and LC-MS/MS for accredited analysis in Slovenia and Austria. For this work, GC-MS/MS analysis was performed in the laboratory of the Federal College and Research Institute for Oenology and Pomology of Klosterneuburg which is accredited to do the official analysis for the Austrian wine industry. Some of the wines of survey 2 were also analysed at the Agricultural Institute of Slovenia using the same method as published in this study. The results were similar.

Line 198. HNO3

This was corrected

Line 211. 1.8 µm

This was corrected

Line 216. Carrier gas?

Thanks for spotting the mistake. We corrected for “collision gas” (line 251)

Line 217, 218. min-1

This was corrected

Line 219. Table X? Corrected:

Table 2. This was corrected

chapter 2.6. You should include what software the boxplots were made in

This was added line 246 in the new manuscript

Table 1. m/z in italic

This was corrected

Figure 1. the two digits after the decimal point on the y-axis can be eliminated

This was corrected in Figure 1 and 4

Lines 243-256. Perhaps it could be useful to contextualize the data with respect to the olfactory threshold?

This was already done, see lines 378-384 in the revised manuscript.

Moreover, the link between 2-AAP concentration and UTA perception is not always clear, the matrix effect can be dramatic. Therefore, it did not seem relevant to us to discuss more this aspect. Further work will have to clarify this sensory aspect of UTA.

Line 287. Was wine A used for experiment 3.2? How were the two wines different? Some details could be added in supplementary

Some information about wine A and B were added in supplementary data 2

Line 418. Metal seizure by lees absorption?

This cannot be excluded even though we showed that Inactivated Yeast Derivates are less efficient than tannin to limit 2-AAP production. It is discussed line 423-430

Reviewer 2 Report

The article "New Insight on the Formation of 2-Aminoacetophenone in White Wines" presents valuable findings on the formation of 2-aminoacetophenone. However, there are a few concerns that should be addressed to enhance the quality and impact of the article.

Introduction: The introduction section should provide a comprehensive background on 2-aminoacetophenone formation in white wines. The authors should discuss previous studies, existing theories, and knowledge gaps in this area. This will help readers understand the significance and novelty of the research presented in the article.

Experimental Details: The article lacks sufficient details on the experimental procedures. It is crucial to provide information about the wine samples, their origins, and the specific analytical techniques employed. Additionally, any relevant controls or replicates should be mentioned. By providing these details, the authors can ensure the reproducibility and reliability of their findings.

many errors in formatting has been observed

authors not follow the scientific writing protocol if sing italics, superscript, subscript

Manuscript needs major revisions

Addressing these concerns will significantly strengthen the article and ensure its contribution to the scientific community

authors not follow the scientific writing protocol if sing italics, superscript, subscript

Language and formatting needs to be thoroughly revised

Author Response

The answers to reviewers are written in green.

The article "New Insight on the Formation of 2-Aminoacetophenone in White Wines" presents valuable findings on the formation of 2-aminoacetophenone. However, there are a few concerns that should be addressed to enhance the quality and impact of the article.

Thanks for your overall positive feedback

Introduction: The introduction section should provide a comprehensive background on 2-aminoacetophenone formation in white wines. The authors should discuss previous studies, existing theories, and knowledge gaps in this area. This will help readers understand the significance and novelty of the research presented in the article.

We agree with reviewer 2 that the novelty of this work should have been better introduced. For that matter, we added some sentences in the revised manuscript (line 85-92).

On the other hand, we have checked carefully the literature again and also how the articles dealing with 2-AAP were introduced and we did not see what else could be added to our introduction. We do not think that the introduction of such an article should be a comprehensive literature review but it should rather give a relevant overview on the topic that will help the reader to understand the context of the study and which novelty it brings.

Experimental Details: The article lacks sufficient details on the experimental procedures. It is crucial to provide information about the wine samples, their origins, and the specific analytical techniques employed. Additionally, any relevant controls or replicates should be mentioned. By providing these details, the authors can ensure the reproducibility and reliability of their findings.

We do not understand well reviewer 2’s comments here. “Material and method” is the largest section of this manuscript. All the analytical methods and procedures were described in details. See from line 93 to 268 in the revised manuscript. We added some information about chemical suppliers (line 94-106) and the wines used for the experiments in supplementary data 1 and 2. Information about replicates were written and can be found in the revised manuscript lines 173-176.

many errors in formatting has been observed

Reviewer 2 is right; we apologize for that and we corrected

authors not follow the scientific writing protocol if sing italics, superscript, subscript

Indeed, thanks for pointing out these mistakes, we corrected them

Manuscript needs major revisions

Addressing these concerns will significantly strengthen the article and ensure its contribution to the scientific community

Round 2

Reviewer 2 Report

Authors have made all the necessary amendments, manuscript is acceptable in its current form.